# A Walkway from Crayfish to Oligochitosan

**Evgeniya A. Bezrodnykh, Oxana V. Vyshivannaya** **, Boris B. Berezin, Inesa V. Blagodatskikh and Vladimir E. Tikhonov ***

A. N. Nesmeyanov Institute of Organoelement Compounds (INEOS), Russian Academy of Sciences, 119991 Moscow, Russia

***** Correspondence: tikhon@ineos.ac.ru

**Abstract:** Edible crayfish are an object of local fishing and artificial breeding in many countries. This industry is very promising in terms of production of healthy foods and byproducts, such as biologically active polyaminosaccharide—chitosan and its derivatives. However, crayfishing is far from being at the level at which it could be. This laboratory scale protocol describes a walkway from crayfish *Actacus leptodactylus* to chitin, chitosan, and oligochitosan hydrochloride, with the main emphasis on the way of getting rid of the impurities (residual heavy metals, proteins and other residues) commonly present in commercial chitosan and its derivatives, as well as the characterization of the products by means of inductively-coupled plasma mass spectrometry (ICP-MS), energy-dispersive X-ray spectroscopy (EDXS), protein and elemental analysis, proton magnetic resonance spectrometry ($^1$H NMR), and chromatography methods. The protocol includes the preparation of crude shell waste; the extraction of proteins from crude shell waste and preparation of deproteinated shell waste, demineralization and decolorization of the deproteinated crayfish shell waste, deacetylation of chitin, and depolymerization of chitosan. EDXS shows the presence of Al and Si residues in chitin is found when the deproteination of crayfish waste is carried out in an alumosilicate glass vessel. In contrast, these residues are absent when deproteination is carried out in the borosilicate glass flask. Analytical data show that the content of residues in chitosan and oligochitosan hydrochloride meets pharmaceutical requirements. The study demonstrates crayfish waste a promising for the purification of chitosan, for the preparation of pharmaceutical grade oligochitosan hydrochloride, and can improve commercialization of crayfishes.

**Keywords:** crayfish; chitin; chitosan; oligochitosan; analysis; proteins; heavy metals

## 1. Introduction

Chitosan, poly[β-(1→4)-2-amino-2-deoxy-D-glucose], is the by-product of the crustacean food processing industry and is mainly produced from crab, shrimp, or krill shells [1]. Low toxicity, biodegradability, and biocompatibility, together with a nonspecific antimicrobial activity has made chitosan a product of worldwide application in pharmaceutical, food, and cosmetic compositions [2–8].

The growing market for chitosan [9] requires the application of pure material free of harmful residual impurities (proteins and heavy metals) as chitosan is used, in particular, as an ingredient in biopharmaceuticals, skincare, and other cosmetic compositions [5,10]. Unfortunately, both chitin that is separated from aquatic crustaceans and chitosan that is industrially manufactured from chitin can contain excessive amount of proteins and residual heavy metals, which come from polluted land-locked seas and rivers [11].

The narrow-toed crayfish *Actacus leptodactylus* is an object of local fishing and artificial breeding in many countries [12]. As with other crustaceans, chitin is the second main component of crayfish shells [13]. Unfortunately, the growing application of chitosan produced from crayfish is far from being at the level it could be at [14,15]. The solution to this could be the use of crayfishing in not only in food processing and improvement of soil structure but also for the extraction of chitin and production of chitosan and its derivatives

as additional products of crayfishing, as this would increase the economic efficiency of crayfish plants by widening the application of crayfish waste and chitosan [16–18].

For several decades, the terms chitin, chitosan, water soluble chitosan, low molecular weight chitosan, oligochitosan, chitosan oligomers, chitosan oligosaccharides, heterochitosans, heterochitosan oligosaccharides, chitin oligomers, and chitooligosaccharide have been widely used in scientific literature. Unfortunately, all these terms are used voluntary since the terminology in chitin/chitosan sciences has not been commonly accepted so far. The terms "chitin" and "chitosan" are the most used terms in chitin and chitosan sciences ("chitinology"). The term "chitin" describes a homopolymer of N-acetyl-β-D-glucosamine linked via 1,4-bonds. Chitin extraction from natural sources usually has molecular weight of a million Daltons and more. Chitin is insoluble in organic (acetic and lactic) acids but is highly swelling and hydrolyzing in concentrated inorganic (hydrochloric and phosphoric) acids.

Deacetylation of chitin by one or another method leads to the preparation of a completely or partially deacetylated chitin. When the degree of deacetylation reaches the value of 60% or more, chitin becomes soluble in diluted acetic or hydrochloric acids. This soluble form of chitin is named "chitosan". Thus, the solubility in aqueous acidic media is the criterion for distinguishing between chitin and chitosan. Here, we should take into account that the term "chitosan" is not a chemical but a conditional one. As shown and discussed in several articles, chitosan could be distinguished from oligochitosans (short chain chitosans) by molecular weight (MW). Still, the boundary between these groups are fluid, although the term "oligochitosan" can be used for chitosan molecules with MW $\leq$ 16 kDa (degree of polymerization $\leq$100) [19–22].

So far, a number of articles published have been devoted to the elaboration of varied methods for the separation of chitin from different chitin containing sources [23]. In spite of the multiplicity of the methods, all of them include the main ones, such as deproteination, demineralization (decalcification), and decolorization (if required). Deproteination is required since crayfish, like other crustacean animals, contains unbound edible proteins and polysaccharide conjugated proteins [13]. While unbound edible proteins are used for human and animal food production, the presence of the remaining conjugated proteins can pose a threat to human health and may also have a bad impact on the stability of a chitin/chitosan based product when chitin and chitosan are used as food additives or as components of pharmaceutical and cosmetic compositions [24].

Demineralization is required since crustacean shells are mineralized and contain (mainly) calcium carbonate and foreign metals in the form of carbonates and phosphates [25]. Although the process of chitin separation from crustacean shell wastes includes a demineralization stage, which eliminates calcium carbonate and most of residual heavy metals from chitin [26], it has been found that commercial chitosan batches can contain an excessive concentration of heavy metals [27,28], which can reduce the quality and shelf-life of chitosan containing products [5].

From here, the main problem arising when preparing pharmaceutical-grade oligochitosan hydrochloride is the presence of harmful residues in the chitosan used for the preparation. Unfortunately, the protocols published so far have not touched the problem of proteins and heavy metals remaining in chitosan separated from crayfish and other crustaceans, and have been focused mainly on the physicochemical properties of chitosan (solubility, solution viscosity, molecular weight, degree of deacetylation) and the presence of ash and residual moisture [13,15,17,29–32]. As to the hydrochloric acid salts of chitosan and its derivatives, there have been two protocols published so far on the preparation of chitosan hydrochloride [33,34] and two protocols devoted to the hydrolysis of commercial chitosan and the preparation of oligochitosan hydrochloride with characteristics corresponding to the requirements of the *European Pharmacopoeia* for chitosan hydrochloride [28,35,36].

In the present study, we describe the extraction of chitin, preparation, and analysis of chitosan and oligochitosan hydrochloride focusing mainly on the purity of chitosan and oligochitosan hydrochloride from residual heavy metals and proteins. The protocol

included: (a) preparation of crude shell waste; (b) extraction of proteins from crude shell waste; (c) demineralization and decolorization of the waste and preparation of chitin; (d) deacetylation of chitin; and (e) prapartion of oligochitosan hydrochloride.

## 2. Materials and Methods

### 2.1. Matherials

Sodium hydroxide (CAS number 1310-73-2), 37% hydrochloric acid (grade: ACS, ISO, Reag. Ph. Eur), 30% hydrogen peroxide (grade: ISO), and Coomassie Brilliant Blue G-250 (CBG250, CAS number 6104-59-2) were purchased from Merck. Deuterium oxide (99.0%) and deuterium chloride (99%) were the products of Aldrich.

Fresh-frozen crayfish *A. leptodactylus* was bought on the local market. All further work with crayfishes was carried out in the accordance with the requirements approved by the Bioethics Committee at the Federal Research Center's Fundamentals of Biotechnology of the Russian Academy of Science (Protocol N°16/2, 21 May 2022) and were performed according to the corresponding guidelines for animal use.

### 2.2. Separation of Chitin

Fresh-frozen crayfishes (4 kg; 7–9 cm in size) were boiled in distilled water for 10 min. Crayfish meat was carefully separated from the exoskeleton mechanically. Crayfish shell waste was passed through a meat grinder to grind it into particles of 1 to 3 mm in size, and then washed with excess of distilled water on a sieve with a pore diameter of 0.3 mm to delete unbound proteins. The washed waste was pressed to delete excess water, and then the wet waste was subjected to deproteinization with 1 M NaOH (4 L) at 90 °C for 4 h with stirring in a glass vessel. After filtration, the solid product was collected and washed with water, and then it was demineralized by 1 M HCl (4 L) while stirring in a glass vessel at room temperature for 6 h. Deminerelized chitin was filtered off, washed with exces of distilled water until the neutrality of effeluent water was reached, and pressed to remove excess water. Chitin was decolorized with 0.25% NaOCl solution (1 L) at room temperature for 5 min. The wet product was dried in a vacuum at 70 °C for 3 h, and then it was milled to powder with particle sizes ≤0.5 mm. The yield of crayfish chitin was 72 g (1.8%) with a humidity of 3.4%.

### 2.3. Separation of Chitosan

Chitin (70 g) was deacetylated with excess of 40% (*w/v*) NaOH solution (500 mL) at 120 °C for 2 h in HK 500 polytetrafluorethylene-made reactor equipped with a stainless steel high pressure chamber. After deacetylation, the product was filtered off and washed with excess of deionized water until neutrality of the effluent water was reached. The wet chitosan was dried in a vacuum (0.01 mPa) at 70 °C for 3 h. The yield of dry crayfish chitosan was 56 g (DD 96%) with a humidity of 3.1%.

### 2.4. Preparation of Oligochitosan Hydrochloride

Oligochitosan was prepared following the protocol published in [35]. Briefly: chitosan (50 g) was dissolved at mixing in 1 M hydrochloric acid (500 mL) containing 1.5% hydrogen peroxide at 70 °C in a Schott Duran glass flask for 2 h, and then the resultant solution was cooled to room temperature and then diluted with ethanol (2.5 L). The precipitate was filtered off, washed with ethanol (100 mL), and dried in a vacuum (0.01 mPa) at 70 °C for 2 h. The yield of oligochitosan hydrochloride was 40 g with a humidity of 9.1%. Elemental analysis: 33.00% C, 6.31% H, 6.55% N, 16.33% Cl.

The degree of chitosan deacetylation (DD ± 1, mol. %) was determined by the $^1$H-NMR method and calculated as discribed in [37] at 20 °C using 1% DCl solution in $D_2O$ as a solvent and tetramethylsilan as an internal standar. The method used the ratio of 1/3 part of the integral intensity of $CH_3$ groups ($I_{CH3}$) of acetyl residues ($CH_3CO$) versus 1/6 part of the sum of integral intensities of H2-H6 protons (4 protons belonging to glucosamine and N-acetylglucosamine residues and 2 protons of $CH_2$ groups), while the C1 proton was

screened under a water signal (H in $DCl/D_2O$ and residual signals from water in chitosan and oligochitosan hydrochloride):

$$DD(\%) = [1 - 2I_{CH3}/I_{H2\text{-}H6}] \times 100 \tag{1}$$

Molecular characteristics (weight average molecular weight $M_w$, number average molecular weight $M_n$, pick molecular weight $M_p$, and polydispersity index $M_w/M_n$) of chitosan and oligochitosan hydrochloride were measured by the HP-SEC method using an Agilent 1200 Series Chromatography system equipped with a RI detector and Ultra-Hydrogel 500 and 250 Å columns using 0.225 M acetic acid/0.3 M ammonium acetate buffer as an eluent. A series of monodispersed pullulans (Fluka) with molecular weights from 1.08 to 710 kDa) were used as calibration standards.

*2.5. Protein Content Analysis*

Protein concentration was determined by the modified colorimetric method described in [38] using CBG250 as a protein dye reagent and BSA as a standard for protein concentration analysis. Briefly: a crude chitin or chitosan sample (1 g) was dispersed in 1 M NaOH solution (75 mL) using a Schott Duran vessel equipped with a cover, and the required amount of 1 M NaOH solution was added to the vessel to reach a total volume of 100 mL. The dispersion was stirred at 95 °C for 1 h and cooled to room temperature. The sample was separated from the supernatant by filtration on a glass filter, and the alkaline effluent was neutralized with concentrated hydrochloric acid to pH $7 \pm 1$. Aliquot volume of solution (diluted if required) was mixed with Bradford's reagent [39]. After centrifugation for 5 min at 3000 rpm, the absorbance of solution was measured at 595 nm. Protein concentration was determined using BSA-calibrated curve with a limit to detection of 1 μg/mL.

The content of residual heavy metals was determined by inductively coupled plasma mass spectrometry (ICP-MS) using a spectrometer ELAN DRC-II ICP (Perkin-Elmer, Waltham, MA, USA).

Solubility, insoluble matter and solution appearance were determined in accordance with the requirements of The European Pharmacopeia, 2004 [36], and The United States Pharmacopeia, 2013 [40].

Carbon, hydrogen, nitrogen, chlorides, ash, and humidity (water content calculated by Fisher's method) were determined by using laboratory facilities for elemental microanalysis of INEOS.

Energy-dispersive X-ray (EDX) analysis for solid phases was performed on Phenom ProX instrument using the standard manufacturer's software package (PhenomWorld, Eindhoven, The Netherlands).

## 3. Results and Discussion
### 3.1. Crayfish Waste Treatment and Separation of Chitin

In the present study, the main attention was devoted to the elaboration of a laboratory protocol for the preparation of oligochitosan hydrochloride directly from the crayfish *Actacus leptodactylus* aqua cultured in an artificial lake, with the purpose of determining how the level of protein and heavy metals in the crayfish chitosan, as a precursor of oligochitosan, met the requirements of USP2013 to become pharmaceutical-grade chitosan. As was shown earlier, commercial chitosan batches contained excessive concentrations of residual heavy metals (mainly Fe, Cr, and Ni) originated from the leaking of these metal ions from stainless steel-made reactors during the process of deacetylation at harsh conditions (high temperature and NaOH concentration) [35]. A lack of data on the applicability of aqua-cultured crayfishes as a source of preparation for pharmaceutically pure oligochitosan hydrochloride moved us to evaluate how *A. leptodactylus* suits to the preparation of oligochitosan hydrochloride suitable to biomedical testing, at least on a laboratory scale.

For this purpose, crayfish shell waste of whole crayfish was separated from the edible crayfish meat. The crude shells, which were disintegrated to tiny pieces, were thoroughly washed to separate most of the unbound proteins. Residual proteins were extracted from the crude shells by 1 M NaOH solution at heating. As shown, the extraction led to a transparent crude extract (CS), brown in color (Figure 1, sample 1). After the deproteination, demineralization, and decolorization of the shells, the residual proteins in the deproteinated shells (DPS) and deproteinated/demineralized/decolorized shells (DDS) were additionally extracted using 1 M NaOH solution to evaluate the content of residual proteins after these stages. As shown in Figure 1 (samples 2 and 3), alkaline DPS extract and DDS extract were yellow in color. Since Branford's reagent is an acidic solution of CBB-250, all alkaline extracts were acidified to pH~7. After the acidification, the CS extract remained brown but become turbid at further acidification to pH < 5 due to precipitation of excessive proteins from the solution (Figure 1, sample 5). In contrast, alkaline DPC and DDC extracts were transparent and light yellow in color (Figure 1, samples 2 and 3) and remained as transparent and almost colorless solutions even after further acidification since they contained less proteins (Figure 1, samples 6 and 7).

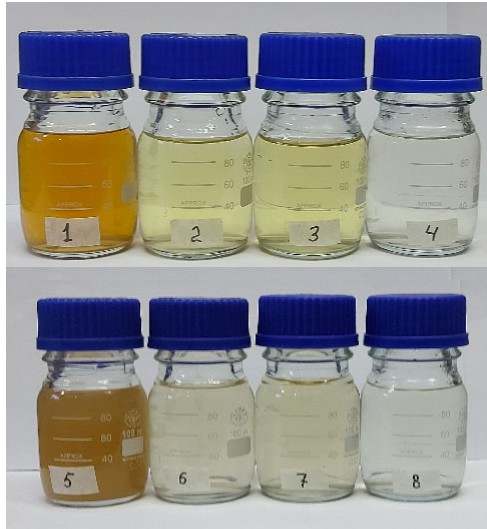

**Figure 1.** Alkaline (1–4) and acidified (5–8) extracts: SC extract (1,5), DPS extract (2,6), DDS extract (3,7), and chitosan extract (4,8).

The concentration of protein in the extracts determined by Bradford's method is shown in Table 1. At this stage, the importance of the glass flask chosen for deproteination and demineralization should be mentioned, namely the glass composition. Preliminary ICP-MS analysis showed the presence of Fe, Cr, and Ni in the crude waste, and the absence of any other heavy metals (Cd, Pb, and Hg) including arsenic (Table 1). After washing off the waste, deproteination, demineralization, and decolorization stages, most of proteins as well as the main part of Fe and all residual Cr and Ni were separated from crayfish chitin (Table 1). Nevertheless, semi quantitative EDX analysis revealed a presence of Al (0.7%) and Si (0.3%) in the chitin (Chitin-AlSiG) collected after the stage when the processes of were carried out in an alumosilicate glass (AlSiG) flask (Figure 2a). In contrast, the use of a borosilicate glass (BSiG) flask, which was more resistant to the alkaline and acidic media, led to the chitin Chitin-BSiG, which contained a low concentration of Fe (Table 1) and residual Si (0.2%) with an absence of aluminum (Figure 2b).

**Table 1.** Content of residual proteins, heavy metals, and arsenic in chitin, chitosan, and oligochitosan. USP 34-NF29 limits: proteins ≤0.2%; Fe ≤ 10 ppm, Cr ≤ 1 ppm, Ni ≤ 1 ppm, Pb ≤ 0.5 ppm, Cd ≤ 0.2 ppm, Hg ≤ 0.2 ppm, and As ≤ 0.5 ppm.

| Compound | Proteins (%) | Fe (ppm) | Cr (ppm) | Ni (ppm) |
|---|---|---|---|---|
| Crude waste | 10 ± 2 | 23.0 | 0.4 | 0.3 |
| Chitin-AlSiG | 2 ± 0.5 | 2.5 | Nf * | Nf * |
| Chitin-BSiG | 2 ± 0.5 | 3.0 | Nf * | Nf * |
| Chitosan | 0.11 ± 0.03 | 3.7 | Nf * | Nf * |
| Oligochitosan | <0.01 | 2.8 | Nf * | Nf * |

* Nf—not found.

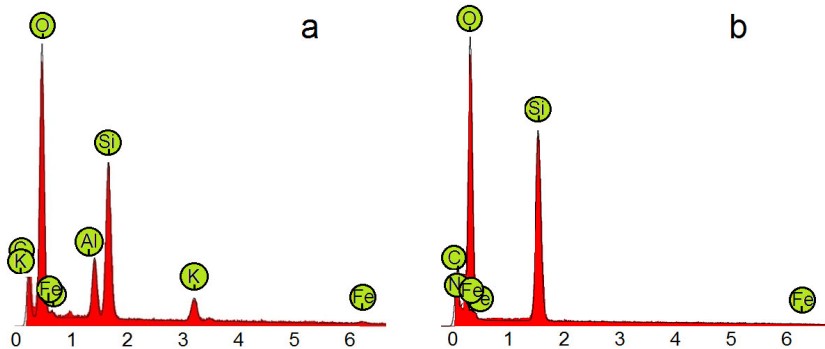

**Figure 2.** (**a**)—EDX spectrum of chitin-AlSiG; (**b**)—EDX spectrum of chitin-BSiG.

### 3.2. Chitin Deacetylation and Preparation of Chitosan

Chitin deacetylation is usually carried out in 30–60% (*w/v*) NaOH solution at an elevated temperature (80–150 °C) and leads to chitosan that is soluble in acidic aqueous media at pH < 6 [1,41]. The origin of chitin plays a significant role in choosing the conditions of chitin deacetylation since the α, β, and γ chitin present in different animals differ in their ability to deacetylate [42]. Depending on the origin of chitin and its concentration and temperature, commercial chitosan obtained has a MW ranged from 20 to 1000 kDa and DD 70–95% [43].

As is mentioned above, the leakage of heavy metals (Fe, Cr, Ni) from stainless steel in harsh alkaline media makes stainless steel reactors unapplicable for the preparation of chitosan, as the content of heavy metals results in chitosan that fails to meet pharmeceutical requirements. For this reason, and for better deproteination and deacetylation, crayfish chitin was deacetyleted by using 40% NaOH (*w/v*) solution at 120 °C and polytetrafluorethylene-made reactor. Like other crusaceans, crayfish shell chitin has an α- structure, which is hard to deacetylate, and requires a high concentration of sodium hydroxide and a high process temperture [29,41].

After deacetylation of chitin-BSiG, ICP-MS analysis of chitosan (Figure 3a) showed a presence of a low concentration of Fe (3.7 ppm). EDX analysis found the rest of the residual Si < 0.1% and a low content of residual proteins in the concentration of 0.11% (Table 1; Figure 1, samples 4 and 8).

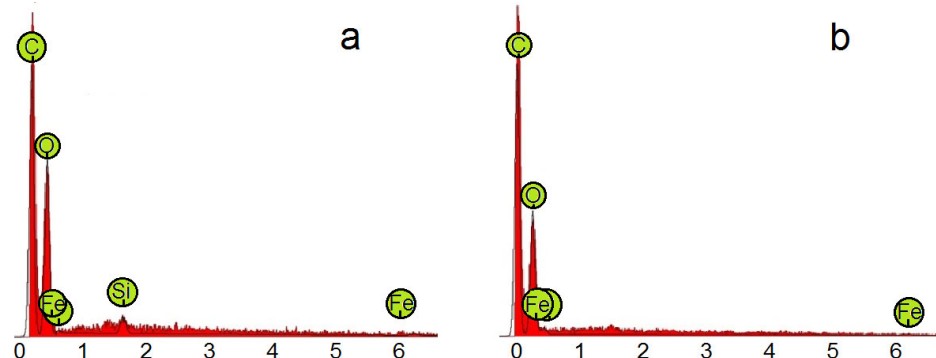

**Figure 3.** (**a**)—EDX analysis of chitosan; (**b**)—EDX spectrum of oligochitosan.

### 3.3. Preparation of Oligochitosan Hydrochloride

As has been shown, the application of oligochitosan instead of chitosan has several advantages due to its lower solution viscosity, higher antimicrobial activity, better mucosal adhesion, and compatibility with neutral and anionic surfactants used in pharmaceutical and cosmetic compositions [20,35,43–47].

Many methods of chitosan depolymerization have been described so far. Among them, acidic and enzymatic, as well as the depolymerization by nitrous acid or by hydrogen peroxide are the main ones. Although all the methods have advantages and disadvantages, a combination of hydrochloric acid and hydrogen peroxide was found as the more efficient and reproducible for the preparation of oligochitosan hydrochloride samples differing in molecular weights, and that reach the physicochemical characteristics meeting pharmaceutical requirements for commercial chitosan contaminated by residual heavy metals [35]. This method was also chosen since it was found to be well reproducible, and it had minimal impact on the chemical structure of oligochitosan in comparison with the method that used hydrogen peroxide or nitrous acid and did not required the application of expensive enzymes whose residues could contaminate the product [48–53]. The method of depolymerization of chitosan in hydrochloric acid/hydrogen peroxide solution was specific towards the cleavage of the bonds between N-acetylglucosamine units [52,54–56].

In this laboratory scale protocol, chitosan was depolymerized with the use of hydrochloric acid and hydrogen peroxide in a borosilicate glass flask, as described earlier in [35]. Following the protocol, the preparation of oligochitosan hydrochloride created a product free of residual proteins (Table 1) with a low content of Fe (2.8 ppm) and the absence of other heavy metals, such as Al and Si residues (Figure 3b).

### 3.4. Molecular Characterization of Chitosan and Oligochitosan Hydrochloride

Chitosan is mainly characterized by molecular weight and degree of deacetylation (DD); i.e., describing chitosan requires that its main characteristics are mentioned. In addition, the term "chitosan" describes a family of polysaccharides that consist of glucosamine and N-acetylglucosamine, and differ by composition and molecular weight distribution. Polydispersity index $D_m = M_w/M_n$, which is equal to the ratio of the weight average molecular weight ($M_w$) and the number average molecular weight ($M_n$), is usually used as a measure of molecular weight inhomogeneity of macromolecules in a polymer sample. All these characteristics determine the physicochemical, antimicrobial, and biological properties of chitosan, as reviewed in [57].

After the separation of chitosan and the preparation of oligochitosan hydrochloride, their molecular characteristics were analyzed by the use of size exclusion chromatography (Figure 4). The calculated values are shown in Table 2.

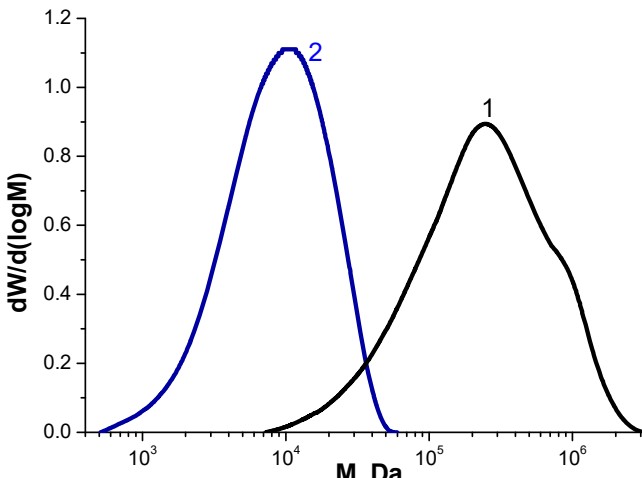

**Figure 4.** HP-SEC elution profiles of chitosan (1) and oligochitosan (2).

**Table 2.** Molecular characteristics of chitosan and oligochitosan.

| Sample | $M_w$ (kDa) | $M_n$ (kDA) | $M_p$ (kDa) | $D_m$ |
|---|---|---|---|---|
| Chitosan | 370 | 128 | 295 | 2.88 |
| Oligochitosan | 11.3 | 6.0 | 8.6 | 1.89 |

As shown, the molecular characteristics of chitosan ($M_w$ 370 kDa; $D_m$ 2.88) and oligochitosan ($M_w$ 11.3 kDa; $D_m$ 2.88) correspond to the values of chitosan prepared from other crayfish species at similar conditions [31] and with the required characteristics of oligochitosan.

The degrees of deacetylation of chitosan and oligochitosan were determined by a simple and fast $^1$HNMR method using the relationship of the integral intensities of the protons belonging to residual acetyl-groups and protons of saccharide subunits. As it was determined, deacetylation of chitin in harsh conditions (40% NaOH solution, 120 °C) led to the product having a high degree of deacetylation of 98% (Figure 5); i.e., the value usually occurred when the deacetylation of crustacean chitin was subjected to deacetyaltion at harsh conditions [31]. Furthermore, the hydrolytic cleavage of chitosan macromolecules accomponed by additional acid deacetylation of chitosan [54] and its product led to oligochitosan with an increased degree of deacetylation of 98% (Figure 5).

As shown in Figure 5, the spectra contains a low signal of residual acetyl-group protons at 1.92 ppm, the signal of proton at C2 position of the saccharide ring at 3.04 ppm, and a complex of signals of C2–C6 protons belonging to glucosamine and N-acetylglucosamine fragments in the range of 3.7–4.0 ppm.

Molecular characteristics of oligochitosan determined by HP-SEC and NMR studies showed that these values were characteristic of oligochitosan and can provide a higher antimicrobial activity of oligochitosan, which grows with increasing DD [20,57]. Just as with the concentration of residual heavy metals and proteins, all other characteritics of chitosan and oligochitosan hydrochloride have been found to be in the range required by the United States Pharmacopeia (2013) for "chitosan" and the European Pharmacopeia (2004) for "chitosan hydrochloride" (Table 3).

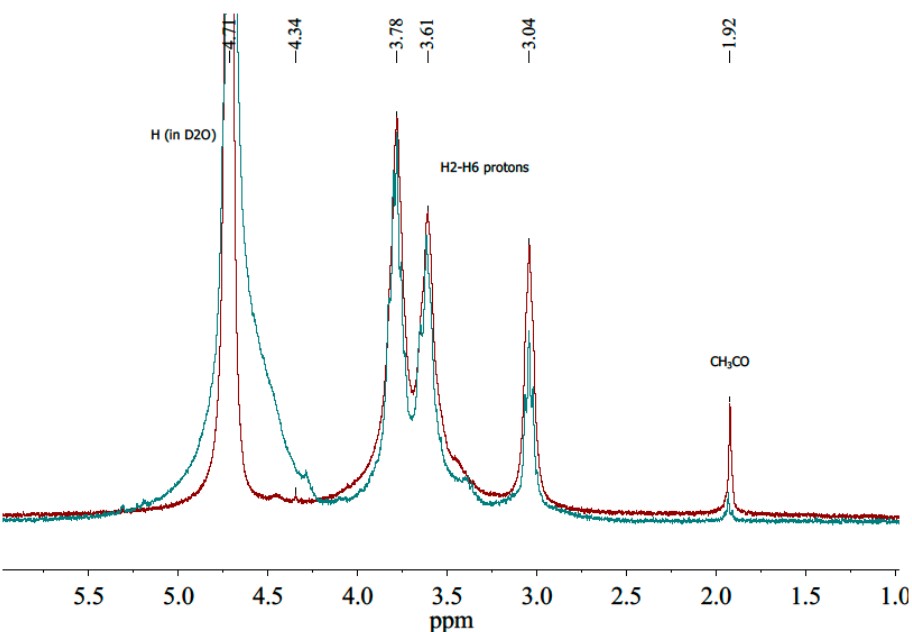

**Figure 5.** $^1$H-NMR spectrum of crayfish chitosan (red) and oligochitosan (green).

**Table 3.** Physicochemical characteristics of chitosan and chitosan hydrochloride.

| Characteristics | USP (2013) Acceptable Criteria for Chitosan | Factual Data for Chitosan | EP (2004) Acceptable Criteria for Chitosan Hydrochloride | Factual Data for Oligochitosan Hydrochloride |
|---|---|---|---|---|
| Appearance (in solid state) | absent | white | white or almost white | white |
| Matter insoluble in water | absent | 0.15% | $\leq$0.5% | 0.05% |
| Degree of deacetylation (%) | 70–95% | 96% | unstandardized value | 98% |
| Proteins (%) | $\leq$0.2% | 0.11 | not applied | <0.01 |
| Chlorides (%) | not applied | not found | 10–20% | 16.3% |
| Humidity (%) | $\leq$5% | 3.1% | $\leq$10% | 9.1% |
| Ash (%) | $\leq$1% | 0.05% | $\leq$1% | <0.01 |

As it has been shown earlier, both chitosan and oligochitosan show both antimicrobial activity and synergistic/additive effects in combination with antibiotics that are promising with regards to their potential as antimicrobial activity-enhancing agents in pharmaceutical compositions [58–61]. These properties of crayfish chitosan and oligochitosan make them promising for the application in pharmaceutical and cosmetic compositions.

## 4. Conclusions

In this study, a walkway from the aqua cultured crayfish *A. leptodactylis* to chitosan and oligochitosan hydrochloride was shown on a laboratory scale. Molecular characteristics of oligochitosan hydrochloride determined by HP-SEC showed acceptable values for oligio-chitosan characteristics. The content of heavy metals and proteins, as well as the physico-chemical properties of chitosan and oligochitosan hydrochloride, were compared with the requirements of the USA and European Union pharmacopoeias. As shown, chitosan and oligochitosan hydrochloride could be prepared as products purified from residual crayfish proteins and heavy metals. It was demonstrated that the use of polytetrafluorethylene-made reactors at the chitin deacetylation stage produced chitosan pure of Fe, Cr, and Ni



residues, which have been usually present in commercial chitosan. In addition, it was shown that the application of an alumosilicate-made vessel used for deproteination of crayfish waste in sodium hydroxide at elevated temperatures should be avoided, since this application led to contamination of chitin by Al and Si residues. The results showed that chitosan extracted from crayfish exoskeletons and oligochitosan hydrochloride could be prepared to qualities meeting the requirements for the permitted presence of residual proteins, heavy metals, solubility, contents of insoluble matter, and other properties. The laboratory protocol might be used for the elaboration of an industrial one that could improve the commercialization of crayfish by widening the utilization of crayfish waste, and for the production of pharmaceutical-grade chitosan and oligochitosan hydrochloride.

**Author Contributions:** E.A.B.: chemical methodology, samples preparation, assisting experiments; O.V.V.: analytical methodology, samples analysis; B.B.B.: samples analysis, software, resources & original graft supporting; I.V.B.: reviewing, supervision & validation; V.E.T.: writing, editing. All authors have read and agreed to the published version of the manuscript.

**Funding:** This work was supported by Russian Science Foundation (grant No. 23-23-00198). NMR, ICP-MS and EDX studies were performed with the financial support from Ministry of Science and Higher Education of the Russian Federation (Contract No. 075-03-2023-642) employing the equipment of the Center for molecular composition studies of INEOS RAS. Elemental analysis and determination ash and humidity were carried out using facilities of the Laboratory for element microanalysis of INEOS RAS.

**Informed Consent Statement:** Not applicable.

**Data Availability Statement:** Data is contained within the article.

**Conflicts of Interest:** The authors declare no conflict of interests.

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
