# Peer review of "A Walkway from Crayfish to Oligochitosan"

_applsci, doi:10.3390/app13053360_

Round 1
Reviewer 1 Report
This a clear, well designed and well presented wok. It merits publication.
Some minor remaks:
i) The authors should comment on the reproducibility of the protocols described
ii) The authors should comment on the feasibility of the protocols for large-scale application.
iii) If possible, t would be nice to see the NMR spectra and the methodology for the calculation of DD, as supplementary material.
iv) The same stands for SEC traces obtained for the molecular weight determination.
Author Response
- i) The authors should comment on the reproducibility of the protocols described; ii) The authors should comment on the feasibility of the protocols for large-scale application.
Answer: the protocol for chitosan and oligochitosan preparations from whole crayfish is elaborated with the main emphasis on the preparation of chitosan and oligochitosan hydrochloride free of the impurities commonly present in commercial chitosan, namely residual heavy metals and proteins. It is based on the earlier published and well reproduced method described in "OLIGOCHITOSAN HYDROCHLORIDE: PREPARATION AND CHARACTERIZATION, INEOS OPEN, 2021, 4 (3), 103–106; DOI: 10.32931/io2111a", and it can be used for elaboration of a large-scale or semi industrial one
iii) If possible, t would be nice to see the NMR spectra and the methodology for the calculation of DD, as supplementary material; +iv) The same stands for SEC traces obtained for the molecular weight determination.
Answer: Accepted. 1H NMR spectra and HP-SEC figure of chitosan and oligochitosan are shown in the revised manuscript.
Dear Reviewer, thank you very much for your interest to our manuscript and valuable remarks.
Dr. Vladimir Tikhonov
Reviewer 2 Report
This manuscript describes a procedure for the extraction and purification of chitin from crayfish Actacus leptodactylus, followed by the preparation of oligochitosan in high purity to be applied in pharmaceutical applications.
The line numbers of the manuscript are not shown, thus making it more difficult to review the manuscript.
1. Introduction: There is a lack of information on the extraction and preparation methods of oligochitosan. What are the extraction and purification methods described in the literature to obtain oligochitosan. What are the main differences between those methods described and this method. This reviewer suggests that sections 3.1 and part of 3.2 should be moved to the introduction.
Section 2.2: Chitin was decolorized with 0.25% NaOCl solution (1 L) at room temperature for 5 min. What was the percentage of chitin used?
Section 2.4_ Degree of chitosan deacetylation (DD±1, mol. %) was determined by 1H-NMR method as discribed in [20].
This method should be briefly described in this section.
Page 5, What is the meaning of DPC and DDC extracts?
Please explain this confusing sentence: Nevertheless, semi quantitative EDX analysis revealed a presence of Al (0.7%) and Si (0.3%) in the chitin (Chitin-AlSiG) collected after the stage when the processes of were carried out in an alumosilicate glass (AlSiG) flask (Fig. 2 a).
It is very interesting the analysis of the reaction vessels contamination in the in the polymer. These results should be highlighted in this work.
Overall , this manuscript can be improved by comparing its results with results already published for the same species of raw material. Furthermore it is not clear what is the novelty of this manuscript, since this method has been already described (although it was described to purify commercial chitosan). Thus, the major objective of this work should be further demonstrated.
This manuscript should be accepted with major revisions.
Author Response
- Introduction:There is a lack of information on the extraction and preparation methods of oligochitosan. What are the extraction and purification methods described in the literature to obtain oligochitosan. What are the main differences between those methods described and this method. This reviewer suggests that sections 3.1 and part of 3.2 should be moved to the introduction.
Answer: So far, no protocols have been published on the preparation of oligochitosan directly from whole crustaceans. The methods for the preparation of oligochitosan from commercial chitosan are mentioned in the revised manuscript. The protocol for chitosan and oligochitosan preparations from crayfish is elaborated with the main emphasis on the preparation of chitosan and oligochitosan hydrochloride free of the impurities commonly present in commercial chitosan, namely residual heavy metals and proteins. Parts 3.1 and 3.2 are added to Introduction.
Section 2.2: Chitin was decolorized with 0.25% NaOCl solution (1 L) at room temperature for 5 min. What was the percentage of chitin used?
Answer: All chitin separated from the crayfish waste was decolorized so that the yield of crayfish chitin was 72 g (1.8%) with humidity of 3.4% as shown in the experimental part. Two (2) grams of chitin were used for analysises, 70 g were used for preparation of chitosan, 50 g of chitosan were used for preparation of oligochitosan.
Section 2.4_ Degree of chitosan deacetylation (DD±1, mol. %) was determined by 1H-NMR method as discribed in [20].This method should be briefly described in this section.
Answer: Accepted. The method for DD calculation was elaborated by Prof. Hirai is reliable and simple. It uses the relation of 1/3 part of the integral intensity of CH3 groups (ICH3) of acetyl residues (CH3CO) versus 1/6 part of the sum of integral intensities of H2-H6 protons (4 protons belonging to glucosamine and N-acetylglucosamine residues and 2 protons of CH2 groups) while C1 proton was screened under water signal (H in DCl/D2O):
DD(%) = [2ICH3/IH2-H6]·100
Proton at C1 is screened under water signal since the samples were dissolved in deuterochloric acid.
Page 5, What is the meaning of DPC and DDC extracts?
Answer: as described, deproteinated cray fish shells (DPS) were subjected to demineralization and delocolorization so that they are named as deproteinated/demineralized/decolorized shells (DDS).
Please explain this confusing sentence: Nevertheless, semi quantitative EDX analysis revealed a presence of Al (0.7%) and Si (0.3%) in the chitin (Chitin-AlSiG) collected after the stage when the processes of were carried out in an alumosilicate glass (AlSiG) flask (Fig. 2 a). It is very interesting the analysis of the reaction vessels contamination in the in the polymer. These results should be highlighted in this work.
Answer: to detect the presence of heavy metals in chitosan and oligochitosan ICP-MS method was used following the protocol required by EU and USA pharmacopoeias and using special standards of Fe, Cr, Ni, Pb, Hg, and As. We tried to used the energy-dispersive X-ray (EDX) analysis as well. Unfortunately, this method is not quantitative and we had no all the standards to be used in this method. Fortunately, we could detect the leakage of Si and Al from the vessels used but the results of EDXS can be considered as qualitative or semi quantitative for these light elements. Dear Reviewer, this manuscript describes our laboratory scaled protocol, where glass vessels were used on the deproteination and demineralization stages.
Overall , this manuscript can be improved by comparing its results with results already published for the same species of raw material. Furthermore it is not clear what is the novelty of this manuscript, since this method has been already described (although it was described to purify commercial chitosan). Thus, the major objective of this work should be further demonstrated.
Answer: Dear Reviewer, no protocols have been published on the preparation of oligochitosan directly from whole crayfish so far. The use of crayfish waste leads to uncertainty since the waste can contain different amount of residual proteins. The economy of crayfishing is based on the fresh-weight of whole crayfishes. Our protocol for chitosan and oligochitosan preparations from whole crayfish is elaborated with the main emphasis on the preparation of chitosan and oligochitosan hydrochloride free of the impurities commonly present in commercial chitosan, namely residual heavy metals and proteins. Your recommendation to revise the Abstract is accepted.
Dear Reviewer, thank you very much for your interest to our manuscript and valuable remarks.
Dr. Vladimir Tikhonov
Reviewer 3 Report
This work suggests a feasible walkway form crayfish to oligochitosan with special emphasis on the preparation of oligochitosan hydrochloride free of the impurities commonly present in commercial chitosan and its derivatives. This work can improve the commercialization of cray-fish by increasing the use of crayfish waste and the production of chitosan and oligochitosan hydro-chloride. However, some important problems need to be considered.
(1) Some spelling mistakes need to be corrected. Such as “at which whey could be”,
(2) Some spectral characterization should be provided as additional evidences, such as PL or Raman spectra.
(3) EDX analysis about chitosan is not accurate. Do you have any other evidences?
(4) How to reduce the residual heavy metals and proteins? Some detailed description need to be considered.
Author Response
(1) Some spelling mistakes need to be corrected. Such as “at which whey could be”,
Answer: Accepted and corrected.
(2) Some spectral characterization should be provided as additional evidences, such as PL or Raman spectra.
Answer: Dear Reviewer, our revised manuscript includes microanalysis, NMR and HP-SEC data of chitosan and oligochitosan obtained. We recorded FTIR spectra of these products as well. No differences were found in FTIR spectrum of the products comparing with that of commercial chitosan. In my opinion as a reviewer of several Elsevier, Wiley, ACS, and MDPI journals, I believe that there is no sense to overload this manuscript by excessive data. Dear Reviewer, I hope you admit my opinion.
(3) EDX analysis about chitosan is not accurate. Do you have any other evidences?
Answer: the protocol for chitosan and oligochitosan preparations from whole crayfish is elaborated with the main emphasis on the preparation of chitosan and oligochitosan hydrochloride free of the impurities commonly present in commercial chitosan, namely residual heavy metals and proteins. To detect the presence of heavy metals in chitosan and oligochitosan ICP-MS method was used following the protocol required by EU and USA pharmacopoeias and using special standards of Fe, Cr, Ni, Pb, Hg, and As. We tried to used the energy-dispersive X-ray (EDX) analysis as well. Unfortunately, this method is not quantitative, and we had no all standards to be used in this method. Fortunately, we could detect the leakage of Si and Al from the vessels used although the results of EDXS can be considered as qualitative for these elements.
(4) How to reduce the residual heavy metals and proteins? Some detailed description need to be considered.
Answer: as described in our manuscript, crayfish grown even in pure water contains heavy metals with the concentration that is twice above the limits required by the EU and USA pharmacopoeias. After the demineralization of chitin and acid hydrolysis of chitosan the concentration falls below the limits. Our protocol describes the stages and the method used to control the content of heavy metals, proteins and other residues in crayfish chitosan and oligochitosan. Other known method to reduce the content of heavy metals in commercial chitosan are mentioned in our cited article " B.B. Berezin et al., Tikhonov, Extraction of residual heavy metals from commercial chitosan and approach to oligochitosan hydrochloride, Carbohydrate Polymers, 2019, 215, 316-321" .
Dear Reviewer, thank you very much for your interest to our manuscript and valuable remarks.
Dr. Vladimir Tikhonov
Reviewer 4 Report
This article provides a method to treat the shell of aquaculture crayfish to prepare chitosan with low content of residual protein and heavy metals
and oligochitosan hydrochloride。 This work is very valuable and has high reference value for researchers in related fields. Before publishing this article, the following modifications need to be made.
1. The title "A walk from crayfish to olivochitosan" is too simple and generalized, and should include some important information.
2. The article should be rearranged. There are some obvious problems in the layout format, such as: (1) in page2, In the present study, we describe the separation and analysis of Chinese separated
..........residual heavy metals and proteins. (2) 2.5.
Molecular characteristics (weight average molecular weight Mw............... 0.225M acetic acid/0.3M ammonium acetate buffer as an eluent. (3) 2.7. Content of residual heavy metals determined by inductively coupled plasma mass........................
2.10. Energy-dispersive X-ray (EDX) analysis for solid phases was performed on Phenom ProX
instrument using standard manufacturer software package (PhenomWorld, The Netherlands).
3. In page 1, introduction section. The author mentioned that "Unfortunately, both chitin separated from aquatic-crustaceans and chitosan-industrialily
manufactured from the chitin can contain excessive amount of proteins and residual
heavy metals, which come from polluted land-locked sea and rivers [11].”。 The authors should give some research and methods that have been reported to improve the purity of chitosan, and comment on the advantages and disadvantages of these methods, and then introduce the necessity of this work to improve the purity of chitosan.
4. In the section of page 8, including. “ In this study, a walkway from the aqua cultured crayfish A. leptodactylis to chitosan
and oligochitosan hydrochloride was shown, and physicochemical properties of both chitosan and oligochitosan hydrochloride were determined and compared with those required by USA and European Union pharmacopoeias. The results showed that chitosan
extracted from the crayfish exoskeleton and oligochitosan hydrochloride were prepared
with the qualities meeteing the requirements for the permitted presence of residual proteins, heavy metals, solubility, content of insoluble matter and other properties.”。 The author needs to explain (or mark) what are the requirements and standards from USA and European Union pharmacopoeias, and the specific value.
Author Response
- The title "A walk from crayfish to olivochitosan" is too simple and generalized, and should include some important information.
Answer: Dear Reviewer, our manuscript describes the protocol for chitosan and oligochitosan preparations from whole crayfishes and it is written with the main emphasis on the preparation of chitosan and oligochitosan hydrochloride free of the impurities commonly present in commercial chitosan, namely residual heavy metals and proteins. No protocol has been published on the preparations from whole crayfish with the emphasizes so far. The title is really simple but it has attracted your eyes to it. I hope, the Title and Abstract will attract the attention of other researchers to the problems described in our manuscript. Nevertheless, I revised the Title as "A walk from whole crayfish to oligochitosan"
- The article should be rearranged. There are some obvious problems in the layout format, such as: (1) in page2, In the present study, we describe the separation and analysis of Chinese separated.........residual heavy metals and proteins. (2) 2.5. Molecular characteristics (weight average molecular weight Mw............... 0.225M acetic acid/0.3M ammonium acetate buffer as an eluent. (3) 2.7. Content of residual heavy metals determined by inductively coupled plasma mass........................2.10. Energy-dispersive X-ray (EDX) analysis for solid phases was performed on Phenom ProX instrument using standard manufacturer software package (PhenomWorld, The Netherlands).
Answer: Sorry and thank you very much. This part is corrected.
- In page 1, introduction section. The author mentioned that "Unfortunately, both chitin separated from aquatic-crustaceans and chitosan-industrially manufactured from the chitin can contain excessive amount of proteins and residual heavy metals, which come from polluted land-locked sea and rivers [11].” The authors should give some research and methods that have been reported to improve the purity of chitosan, and comment on the advantages and disadvantages of these methods, and then introduce the necessity of this work to improve the purity of chitosan.
Answer: as described in our manuscript, crayfish grown even in pure water contains heavy metals with the concentration that is twice above the limits required by the EU and USA pharmacopoeias. After the demineralization of chitin and acid hydrolysis of chitosan the concentration falls below the limits. Our protocol describes the stages and the method used to control the content of heavy metals, proteins and other residues in crayfish chitosan and oligochitosan. Another known method to reduce the content of heavy metals in commercial chitosan is mentioned in our manuscript and cited in [ " B.B. Berezin et al., Tikhonov, Extraction of residual heavy metals from commercial chitosan and approach to oligochitosan hydrochloride, Carbohydrate Polymers, 2019, 215, 316-321"] . No protocols for the preparation of chitosan and oligochitosan hydrochloride free of the impurities commonly present in commercial chitosan or chitosan separated from crustaceans have been found. We worked with whole crayfishes since the application of crayfish waste usually used in other publications for separation of chitin and chitosan could contain different amounts of residual proteins and could insert an uncertainty to mass balance and yield calculation. It should be mentioned that the economic calculation of crayfishing is based on the application of fresh-weight of whole crayfishes. I hope, this manuscript will attract an interest of other researchers working with chitin and chitosan.
- In the section of page 8, including. “ In this study, a walkway from the aqua cultured crayfish A. leptodactylis to chitosan and oligochitosan hydrochloride was shown, and physicochemical properties of both chitosan and oligochitosan hydrochloride were determined and compared with those required by USA and European Union pharmacopoeias. The results showed that chitosan extracted from the crayfish exoskeleton and oligochitosan hydrochloride were prepared with the qualities meeting the requirements for the permitted presence of residual proteins, heavy metals, solubility, content of insoluble matter and other properties.”。The author needs to explain (or mark) what are the requirements and standards from USA and European Union pharmacopoeias, and the specific value.
Answer: as required by USP 34-NF29, the limits are: proteins ≤0.2%; Fe≤10 ppm, Cr ≤1 ppm, Ni≤1ppm, Pb≤0.5ppm, Cd≤0.2 ppm, Hg≤0.2 ppm, and As ≤ 0.5 ppm. These data are shown in Table 1. The content of proteins in chitosan and oligochitosan is shown in Table 1 as well.
Dear Reviewer, thank you very much for your interest to our manuscript and valuable remarks.
Dr. Vladimir Tikhonov
Round 2
Reviewer 2 Report
During the revision process the authors addressed the points mentioned by the reviewers. For this reason the paper should be accepted for publication in Applied Sciences without further changes.
Reviewer 4 Report
The author has answered my comments well, and I think this paper can be published now.